# Does Cataract Extraction Significantly Affect Intraocular Pressure of Glaucomatous/Hypertensive Eyes? Meta-Analysis of Literature

**DOI:** 10.3390/jcm13020508

**Published:** 2024-01-16

**Authors:** Andrea Pasquali, Luigi Varano, Nicola Ungaro, Viola Tagliavini, Paolo Mora, Matteo Goldoni, Stefano Gandolfi

**Affiliations:** 1Eye Clinic, University Hospital of Parma, 43126 Parma, Italy; lvarano@ao.pr.it (L.V.); nungaro@ao.pr.it (N.U.); vtagliavini@ao.pr.it (V.T.); paolo.mora@unipr.it (P.M.); stefano.gandolfi@unipr.it (S.G.); 2Department of Physics, University of Parma, 43126 Parma, Italy; matteo.goldoni@unipr.it

**Keywords:** glaucoma, phacoemulsification, ocular hypertension, pseudoexfoliation, open angle glaucoma, angle closure glaucoma

## Abstract

Background and Objectives: This study aimed to evaluate the effect of cataract extraction on intraocular pressure at 6, 12, and 24 months and their difference compared to the baseline in diverse glaucoma subtypes. Materials and Methods: We carried out research in the MEDLINE, Cochrane Library and EMBASE databases, as of April 2022 for relevant papers, filtered according to established inclusion and exclusion criteria. The meta-analysis evaluated the Mean Reduction and relative Standard Error in these subpopulations at predetermined times. A total of 41 groups (2302 eyes) were included in the systematic review. Due to the significant heterogeneity, they were analysed through a Random Effects Model. Results: We obtained these differences from baseline: (1) Open Angle Glaucoma at 6, 12 and 24 months, respectively: −2.44 mmHg, −2.71 mmHg and −3.13 mmHg; (2) Angle Closure Glaucoma at 6, 12 and 24 months, respectively: −6.81 mmHg, −7.03 mmHg and −6.52 mmHg; (3) Pseudoexfoliation Glaucoma at 12 months: −5.30 mmHg; (4) Ocular Hypertension at 24 months: −2.27 mmHg. Conclusions: Despite a certain variability, the reduction in ocular pressure was statistically significant at 6, 12 and 24 months in both Open Angle Glaucoma and Angle Closure Glaucoma, the latter being superior. Data for Pseudoexfoliation Glaucoma and for Ocular Hypertension are available, respectively, only at 12 months and at 24 months, both being significant.

## 1. Introduction

Over 20 years ago, by common accord between the World Health Organization and the International Agency for Prevention of Blindness, the initiative “Vision 2020: The Right to Sight” was created, the results of which were recently published and reveal that, in adults aged 50 or over, the two main causes of blindness or moderate to severe reduction in visual acuity are, respectively, Cataract (45.4%) and Glaucoma (11%) [1].

These are two diseases which often coexist in a single patient and influence one another; the development and progression of a cataract can contribute to alterations in the drainage of aqueous humour while traditional filtering surgeries used to treat glaucoma can lead to the formation or worsening of the opacification in phakic patients [2].

Cataract extraction surgery is one of the best in terms of cost/benefit analysis and still is the most performed surgical procedure every year in several countries [3].

The objective of this meta-analysis is to demonstrate the correlation between cataract surgery and the evolution of the most crucial and the only modifiable factor in glaucoma, Intraocular Pressure (IOP).

## 2. Materials and Methods

A total of 34 studies, including 39 groups, from 1995 to 2019, were included in the meta-analysis (Table 1) based on the following inclusion and exclusion criteria.

Inclusion criteria

Studies providing data on IOP pre-phacoemulsification and post-phacoemulsification.Studies approved by an institutional revision group or by an ethical committee.

Exclusion criteria

Papers not available in English.Papers not available in a digital format.Results on a non-human population.Studies conducted on patients under 18 years of age.Preceding or concurrent trabeculectomy, other major ocular surgery or relevant illness.A follow-up period of less than 12 months.Relevant study arm with less than 15 eyes analysed.Studies on MIGS without an arm treated only with phacoemulsification.Different subtypes of glaucoma included in the same arm.

The result of primary interest in our meta-analysis was the variation in IOP at times t1 (6 months), t2 (12 months) and t3 (24 months) compared to t0 (pre-surgical baseline).

### 2.1. Literature Research Method

We collected publications on the effects of cataract surgery in patients with glaucoma or ocular hypertension via a thorough research on the MEDLINE, Cochrane Library and EMBASE databases up to 1 April 2021.

This systematic review was not registered in the international prospective register of systematic reviews (PROSPERO). In conjunction, we picked the search keywords and inclusion/exclusion criteria based on systematic reviews and previous meta-analyses. Our updated search includes relevant studies not included in previous review works. The keywords used in the research were (“Glaucoma, Open-Angle” OR “Glaucoma, Angle-Closure” OR glaucoma) AND (“Phacoemulsification” OR “phacoemulsification”) AND (“Intraocular Pressure” OR “intraocular pressure”) AND (“Ocular Hypertension” OR “ocular hypertension”) AND (“Glaucoma, Pseudoexfoliative”) AND (“Cataract Extraction” OR “cataract extraction”) AND (“Cataract Surgery” OR “cataract surgery”).

Results were initially selected depending on the relevance of the title and abstract and subsequently on the adherence to the inclusion/exclusion criteria.

The study arms were divided in Primary Open Angle Glaucoma (POAG), Angle-Closure Glaucoma (ACG), Pseudo-Exfoliative Glaucoma (PXG), and Ocular Hypertension (OH). Some authors included, in the same arm, POAG and OH, or POAG and PXG, or ACG and angle closure disease (ACD), where ACD does not imply the presence of glaucomatous damage to the optic nerve. As a result of the impossibility of separating the sub-groups, these were intentionally omitted from the studies selected for our analysis.

In cases where the studies reported and additional treatment arm, e.g., MIGS combined with cataract extraction surgery, only the phacoemulsification arm was included in this study. Certain studies reported different glaucoma subtypes within the same group and were thus excluded from the analysis. Other studies were excluded due to being extrapolated from the same dataset of previously published papers, thus risking that redundant data would be given to the statistical software to be analysed.

During the literature research, we also found that some studies that did not exclude eyes which underwent, prior to or during the follow-up period, to ocular surgeries or laser procedures (e.g., Selective Laser Trabeculoplasty) and were thus ruled out.

The result of this literature review, as represented in the flow diagram (Figure 1), is a total of 39 groups (20 POAG, 11 ACG, 4 PXG e 4 OH) that were selected from the 34 studies. In total, 11 of the selected arms were retrospective in nature, while 28 were prospective. Overall, 2227 eyes were analysed at the start of the follow-up.

Some studies included a wash-out period as part of their methodology, i.e., the IOP levels were measured after a temporary suspension period of glaucoma medication.

Since each study used a different wash-out protocol, when post-wash-out IOP was measured at baseline and in the various follow-ups, these data were used in the analysis; when impossible, IOP values during pharmacological treatment were used instead.

Data were extracted manually from the chosen studies. All selected papers underwent a quality verification process using the SIGN checklist (Scottish Intercollegiate Guidelines Network).

### 2.2. Analysis and Data Synthesis

A meta-analysis is a quantitative technique allowing, from a clinical and statistical standpoint, for researchers to combine different studies on a single clinical issue to reach a single conclusive result with a more consistent statistical power.

The meta-analysis is completed through Statsdirect 3.2 (StaTsdirect, Wirral, UK). The data we focus on are average IOP pre and post surgery and their standard deviation. In the studies where the confidence interval is not reported, the standard error and/or *p*-values are extracted to calculate average and standard error of the difference compared to baseline (t0).

Using a random effects model, we can attempt to estimate the average of the true effect’s distribution. Bigger studies can then lead to more accurate estimates compared to smaller studies, but all effect sizes are included when estimating the average. The assigned weights under a random effects model are more balanced than under the fixed effects model, as larger studies are less likely to dominate the analysis and smaller studies are less likely to be overshadowed.

The pooled difference allows us estimation of the common difference by assuming that all different populations have the same variance. The Z test enables us to evaluate a hypothesis if the population variance is known or the sample size is ≥30.

Non-combinability is checked through the Cochran Q test to verify whether the treatments have the same effect. Heterogeneity is calculated through the I^2^ test and is significant for values between 75% and 100%, working in favour of analysing the meta-analysis through the random effects model.

The DerSimonian and Laird random effects model has a starting point of considering the effects of the studies as different but related. It is chosen because, by accounting for the variance in effect size, it allows us the use of this information and the results to make inferences on how operating to reduce IOP can benefit other glaucoma patients, as per the hypothesis.

The bias indicators used are the Begg and Mazundar rank correlation, a test estimating the correlation between the degrees of the effect size and the degrees of its variance, as well as the Egger regression, which outputs the degree of asymmetry in the funnel plot graphically represented in the Bias Assessment Plot.

Data summarised in the meta-analysis are graphically reported via the random effects model through a forest plot.

## 3. Results

Table 2 collects all studies satisfying the inclusion/exclusion criteria from the search of the databases with their respective average differences compared to baseline with relative standard deviation at 6, 12, and 24 months.

The following results emerged from the analyses of the four sub-groups (POAG (Table A1), ACG (Table A2), PXG (Table A3) and OH (Table A4)).

The results show a significant reduction in IOP after cataract extraction surgery at all three time-points of 6, 12, and 24 months for POAG. Given the high heterogeneity of the included studies in this subgroup at any timepoint (respectively, I^2^ = 78%, I^2^ = 94% and I^2^ = 94.7%), in the analysis of IOP reduction in POAG, a random effects model (DerSimonian & Laird) was applied. The average combined difference was 2.44 mmHg at 6 months, 2.71 mmHg at 12 months, and 3.13 mmHg at 24 months (95% CI). The Z-tests of the differences all had *p*-values of less than 0.0001, indicating significance (Figure 2).

The forest plots demonstrate that the data sets are of good quality and have no publication bias, as evidenced by the symmetry in the funnel plots (Figure A1). The grey boxes in the forest plots represent the effect size of single studies, and the grey rhombus which represents a combined difference greater than one indicates a significant association. The same holds true, at identical timepoints, for angle closure glaucoma. The analysis shows that, after phacoemulsification, there is a significant reduction in IOP in ACG patients at 6, 12 and 24 months from baseline (Figure 2). The results are based on nine included studies for the 6- and 12-month analysis and eight included studies for the 24-month analysis. The high heterogeneity of the included studies was, once again, accounted for by using the DerSimonian and Laird Random Effects Model. The average combined difference in IOP reduction was 6.81 mmHg (95% CI = 4.06 to 9.55) at 6 months, 7.03 mmHg (95% CI = 4.26 to 9.81) at 12 months and 6.52 mmHg (95% CI = 3.84 to 9.21) at 24 months. The Z-test results indicated that the reduction, with a *p*-value of less than 0.0001, was significant (Figure 3).

The forest plot and funnel plot (Figure A2) analysis also showed that the ACG data set is of good quality, and there was no evidence of publication bias, as confirmed by the Begg and Mazundar rank correlation and Egger regression results.

Despite the scarcity of eligible studies, even in the two subgroups, PXG and OH, the results showed that the reduction in IOP was significant at 12 months from baseline in PXG with a combined difference of 5.30 mmHg (95% CI = 2.216671 to 8.375508), and significant at 24 months from baseline in OH, with a combined difference of 2.27 mmHg (95% CI = 0.106467 to 4.444148) (Figure 4).

The random effects model was also applied in these subgroups due to the high heterogeneity, and the forest plot and funnel plot (Figure A3) suggest that no publication bias was detected.

## 4. Discussion

The systematic literature review and subsequent meta-analysis were executed to analyse data already available in the literature relating to the isolated procedure of cataract extraction with an IOL implant on pre-surgical IOP in patients with POAG, ACG, PXG and OH.

As evident in Figure 5, the highest reduction in IOP, in each of the analysed t_x_, is always in Angle-Closure Glaucoma, with values of 6.81 mmHg, 7.03 mmHg e 6.52 mmHg. Inversely, the lowest reduction, in absolute terms, is that of Ocular Hypertension, with a value of 2.28 mmHg at 24 months from surgery.

Various theories were hypothesised in trying to explain these reductions in ocular hypertension:The molecular theory based on the effects on the pattern of the trabecular meshwork: the inflammatory reaction, consequent from surgery, could lead to hyposecretion of aqueous humour, a reduction in resistance to outflow and biochemical alterations in the blood–aqueous barrier.The physiologic theory based on the effects on the ciliary body: it appears that cataract extraction has a relevant effect on the dynamic involving the ciliary body by reducing its anteposition, especially relevant in ACG.The biomechanical theory based on the anatomical changes in the anterior segment: with an improvement in predictive anatomical parameters on the reduction in IOP in an OCT scan, mainly in the aperture of the camerular angle;The biomechanical theory based on the position of the lens: since an excessively anterior position favours the formation of a higher pressure gradient, which can lead to relative pupillary block.The biomechanical theory based on fluid dynamics: the high flow generated from phacoemulsification, in this limited anatomical space, can clean the pattern of the trabecular meshwork and favour the action of the macrophages in that location [38].

Numerous studies have now shown that cataract extraction, significant in the visual field, can lead to an improvement in sight with an improvement in associated quality of life, especially in patients with pre-existing damage to the visual field, e.g., glaucoma patients. These benefits need to be attentively discussed with glaucoma patients, explaining accurately the involved risks and the implications of surgery [39].

The correlation between cataract surgery and glaucoma comprises numerous facets but this only helps in understanding the new possible role that this surgical procedure has in terms of addressing glaucoma. Cataract extraction alone can help in reducing IOP, in selected patients, trying to improve their visual ability, in the short and long term [38].

Preceding meta-analyses, such as Masis Solano et al. from 2018 [40], have already found significant IOP reduction compared to baseline at the end of the follow-ups: 2.7 mmHg for POAG, 6.4 mmHg for ACG, and 5.8 mmHg for PXG.

With this paper, we instead wanted to offer an indication of how much IOP is lowered over time, to help ophthalmologists understand whether, with cataract surgery alone, it is possible to have a reduction such as to reach the target IOP at a pre-established time interval and to maintain it in the medium term.

It is notable that in the Masis Solano study [40], despite the commonly held idea that phacoemulsification on its own is not an effective POAG treatment, it was already stated that, albeit modest compared to IOP reduction in ACG, cataract extraction is a possible alternative in lighter cases in which the safety of the procedure is the main concern. In cases of ACG, the reduction is significant enough and the operation can be considered a first course of action [41].

As reported in the Ocular Hypertension Treatment Study, this surgical procedure safely reduces IOP even in cases of simple ocular hypertension, but it cannot be confidently said that it also reduces the risk of developing glaucoma [16].

Although this analysis has significant results, its limitations must be acknowledged. Data of studies that contemplated therapeutic washout, often with different protocols in performing it, and of other studies in which this was not performed were aggregated. Further work comparing the former with the latter would certainly be interesting to evaluate the weight of this protocol on the effect of cataract surgery in managing IOP, when not influenced by pharmacological therapy.

Among the variables to consider, the surgical technique for cataract extraction deserves further investigation, as phacoemulsification is not available in all areas of the world. However, as reported by Sengupta et al., it appears that the reduction in IOP using Manual Small Incision Cataract Surgery (MSICS) is comparable at 6 months post surgery [42].

Another aspect that certainly deserves evaluation is how treatment is modified over time after phacoemulsification, drawing on this to better understand the extent of the individual active ingredients’ effect in the post-operative period.

## 5. Conclusions

Although additional research is needed to delve into the individual mechanisms and variables of IOP reduction, there are benefits of cataract phacoemulsification in patients with glaucoma and ocular hypertension, not only shortly after surgery, but also in the following years and over the long term in managing IOP.

The effect is surely striking in Angle-Closure Glaucoma but is not to be underestimated in Primary Open Angle Glaucoma and in its Pseudo-Exfoliative subtype, where it is even more apparent.

It is also important to note how, even in patients with just Ocular Hypertension, a non-insignificant reduction in IOP can be found, useful in protecting from the damage it could cause if it were to evolve into a glaucoma even though we cannot be sure to which degree the eventual perimetrical damage evolution would be affected by this surgery, neither in OH nor in manifest glaucoma.

The starting point of this paper was demonstrating the correlation between cataract surgery and the main modifiable factor in glaucoma, intraocular pressure.

Having shown the existence of this correlation, with phacoemulsification reducing IOP, as well as its statistical significance in all subgroups, we suggest inserting this procedure in the therapeutic framework for other subgroups, as it is already the case for ACG. We do nonetheless fully acknowledge the risks, potentially even catastrophic, it presents, but our findings suggest it should be considered to help reach the target IOP value for a patient’s eye.

In conclusion, we suggest considering phacoemulsification not only as a treatment for cataracts, but also to help in managing glaucoma and OH patients.

## Figures and Tables

**Figure 1 jcm-13-00508-f001:**
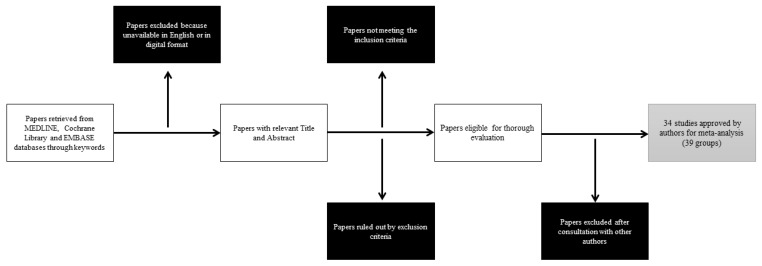
Literature research flow diagram.

**Figure 2 jcm-13-00508-f002:**
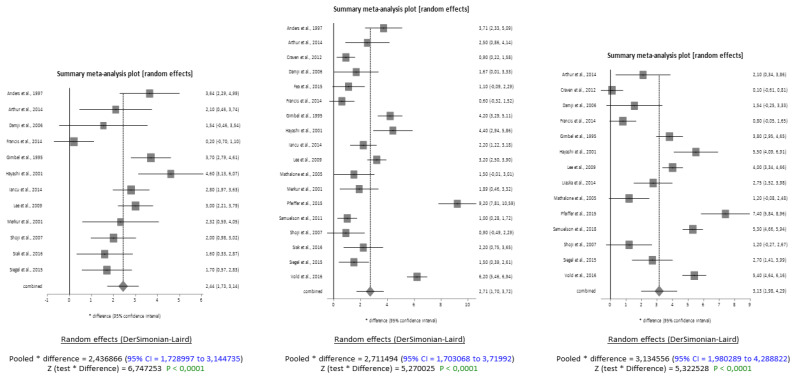
Forest plots meta-analysis on POAG at 6, 12 and 24 months from surgery [7,9,11,19,20,21,22,23,24,25,26,27,28,29,30,31,32,33,34,35].

**Figure 3 jcm-13-00508-f003:**
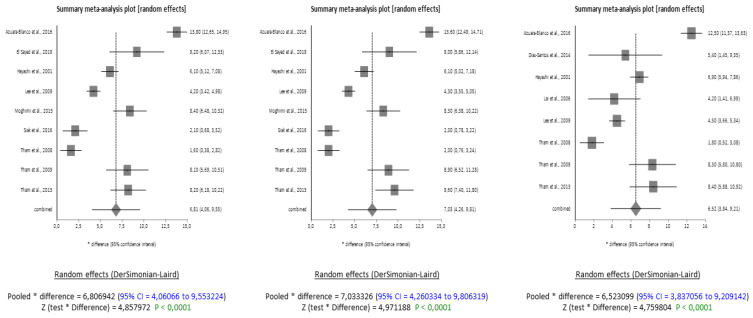
Forest plots meta-analysis on ACG at 6, 12 and 24 months from surgery [4,5,6,7,8,9,10,11,12,13,14].

**Figure 4 jcm-13-00508-f004:**
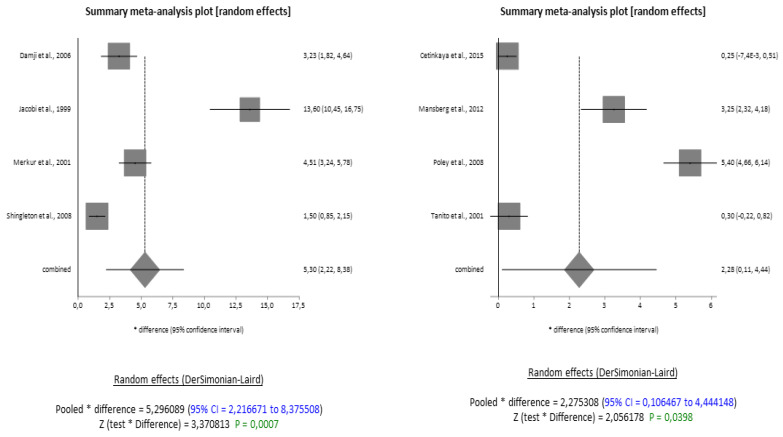
Forest plots meta-analysis on PXG at 12 months and on OH at 24 months from surgery [15,16,17,18,22,29,36,37].

**Figure 5 jcm-13-00508-f005:**
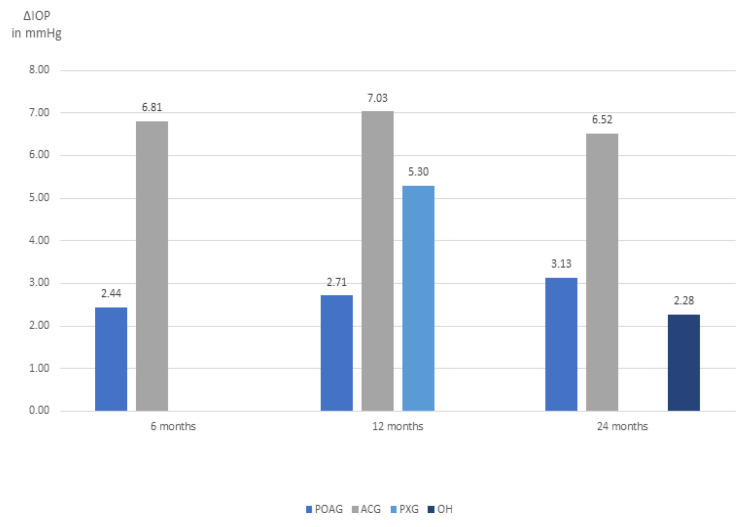
Bar chart of average IOP reduction in mmHg in the 4 subgroups at 6, 12 and 24 months.

**Table 1 jcm-13-00508-t001:** List of studies satisfying the inclusion and exclusion criteria and their characteristics.

Paper	Year	N° of Eyes	Study Design	Glaucoma Subtype	Follow-Up (Months)	Mean and SD Age (Years)	Mean and SD IOP Preop (mmHg)	Mean and SD IOP at 6 Months (mmHg)	Mean and SD IOP at 12 Months (mmHg)	Mean and SD IOP at 24 Months (mmHg)
Azuara-Blanco et al. [4]	2016	208	Prospective	Angle Closure Glaucoma	36	67.0	29.5 ± 8.2	15.7 ± 4.3	15.9 ± 3.2	17.0 ± 3.9
Dias-Santos et al. [5]	2014	15	Prospective	Angle Closure Glaucoma	31.13 ± 4.97	69.5 ± 11.34	19.93 ± 8.30			14.53 ± 1.51
El Sayed et al. [6]	2019	32	Prospective	Angle Closure Glaucoma	20.4 ± 6.5	58.8 ± 8.4	21.6 ± 9.2	12.4 ± 2.5	12.6 ± 2.6	
Hayashi et al. [7]	2001	74	Prospective	Angle Closure Glaucoma	25.7 ± 8.5	73.4 ± 7.3	21.4 ± 3.9	15.3 ± 2.8	15.3 ± 3.6	14.5 ± 2.6
Lai et al. [8]	2006	21	Prospective	Angle Closure Glaucoma	20.7 ± 3.6	73.7 ± 8.1	19.7 ± 6.1			15.5 ± 3.9
Lee et al. [9]	2009	48	Retrospective	Angle Closure Glaucoma	31.1	67.2 ± 6.7	19.5 ± 2.5	15.3 ± 1.8	15.2 ± 1.5	15.0 ± 2.2
Moghimi et al. [10]	2015	46	Prospective	Angle Closure Glaucoma	12	63.2 ± 6.9	22.3 ± 6.3	13.9 ± 3.7	14.0 ± 3.7	
Siak et al. [11]	2016	24	Prospective	Angle Closure Glaucoma	12	70.6 ± 5.5	16.4 ± 4.0	14.3 ± 3.5	14.4 ± 3.5	
Tham et al. [12]	2008	35	Prospective	Angle Closure Glaucoma	24	71.9 ± 6.7	16.3 ± 3.0	14.7 ± 2.8	14.3 ± 2.9	14.5 ± 3.1
Tham et al. [13]	2009	27	Prospective	Angle Closure Glaucoma	24	70.3 ± 7.4	24.4 ± 6.1	16.3 ± 3.5	15.5 ± 3.2	16.1 ± 4.1
Tham et al. [14]	2013	26	Prospective	Angle Closure Glaucoma	24	66.4 ± 8.1	24.1 ± 4.1	15.9 ± 4.2	14.5 ± 4.9	15.7 ± 6.0
Cetinkaya et al. [15]	2015	112	Prospective	Ocular Hypertension	24	61.32 ± 11.12	24.67 ± 2.14	21.00 ± 1.76	23.71 ± 1.11	24.42 ± 1.85
Mansberg et al. [16]	2012	63	Retrospective	Ocular Hypertension	72	64.1 ± 8.9	23.9 ± 3.2	20.17 ± 3.89	20.42 ± 3.61	20. 65 ± 2.69
Poley et al. [17]	2008	81	Retrospective	Ocular Hypertension	49.2 ± 30	70.5 ± 7.4	21.7 ± 2.0			16.3 ± 3.2
Tanito et al. [18]	2001	36	Prospective	Ocular Hypertension	20.5 ± 1.95	72.6 ± 7.9	20.6 ± 1.7	18.9 ± 2.4	19.6 ± 1.9	20.3 ± 1.2
Anders et al. [19]	1997	42	Prospective	Open Angle Glaucoma	12	74.9 ± 9.6	24.71 ± 3.38	21.07 ± 3.68	21.0 ± 3.8	
Arthur et al. [20]	2014	37	Retrospective	Open Angle Glaucoma	21.8 ± 10.1	74.7 ± 9.8	16.2 ± 4.6	14.1 ± 3.3	13.7 ± 3.3	14.1 ± 4.0
Craven et al. [21]	2012	123	Prospective	Open Angle Glaucoma	24	-	17.9 ± 3.0		17.0 ± 3.1	17.8 ± 3.3
Damji et al. [22]	2006	29	Prospective	Open Angle Glaucoma	24	73.99 ± 10.78	18.52 ± 3.52	16.98 ± 4.97	16.85 ± 3.67	16.98 ± 4.21
Fea et al. [23]	2015	24	Prospective	Open Angle Glaucoma	48	-	16.7 ± 3.0		15.6 ± 1.1	
Francis et al. [24]	2014	80	Prospective	Open Angle Glaucoma	36	69.7 ± 6.9	18.1 ± 3.0	17.9 ± 3.5	17.5 ± 3.6	17.3 ± 3.2
Gimbel et al. [25]	1995	53	Prospective	Open Angle Glaucoma	24	77.5	19.3 ± 2.4	15.6 ± 2.9	15.1 ± 2.9	15.5 ± 2.6
Hayashi et al. [7]	2001	68	Prospective	Open Angle Glaucoma	24.1 ± 9.8	73.5 ± 7.9	20.7 ± 5.4	16.1 ± 4.3	16.3 ± 4.2	15.2 ± 3.8
Iancu et al. [26]	2014	38	Prospective	Open Angle Glaucoma	12	71.7 ± 8.27	23.8 ± 2.32	21 ± 2.1	21.6 ± 2.4	
Lee et al. [9]	2009	48	Retrospective	Open Angle Glaucoma	30.8	64.5 ± 9.3	19.1 ± 2.1	16.1 ± 2.3	15.9 ± 1.8	15.1 ± 1.5
Liaska et al. [27]	2014	31	Prospective	Open Angle Glaucoma	24	78.1 ± 7.26	16.65 ± 2.83			13.9 ± 2.7
Mathalone et al. [28]	2005	58	Retrospective	Open Angle Glaucoma	24	78.1 ± 5.7	16.3 ± 4.5		14.8 ± 2.5	15.1 ± 3.2
Merkur et al. [29]	2001	23	Retrospective	Open Angle Glaucoma	18	78.13 ± 6.84	17.22 ± 3.19	14.90 ± 3.51	15.33 ± 2.24	
Pfeiffer et al. [30]	2015	50	Prospective	Open Angle Glaucoma	24	71.5 ± 6.9	26.6 ± 4.2		17.4 ± 3.7	19.2 ± 4.7
Samuelson et al. [31]	2011	117	Prospective	Open Angle Glaucoma	12	73	18.0 ± 3.0		17.0 ± 3.24	
Samuelson et al. [32]	2018	187	Prospective	Open Angle Glaucoma	24	71.2 ± 7.6	18.1 ± 3.1			12.8 ± 3.9
Shoji et al. [33]	2007	35	Retrospective	Open Angle Glaucoma	34.9 ± 19.8	74.9 ± 7.0	16.7 ± 1.4	14.7 ± 2.1	15.8 ± 2.7	15.5 ± 2.5
Siak et al. [11]	2016	30	Prospective	Open Angle Glaucoma	12	67.6 ± 8.1	16.5 ± 4.1	14.9 ± 2.0	14.3 ± 2.4	
Siegel et al. [34]	2015	52	Retrospective	Open Angle Glaucoma	36	78.0 ± 8.1	17.7 ± 4.4	16.0 ± 3.3	16.2 ± 3.4	14.1 ± 2.9
Vold et al. [35]	2016	131	Prospective	Open Angle Glaucoma	24	70 ± 8	24.5 ± 3.0		18.3 ± 3.8	19.1 ± 3.9
Damji et al. [22]	2006	29	Prospective	Pseudoexfoliation Glaucoma	24	72.49 ± 6.32	19.81 ± 2.9	15.73 ± 2.97	16.58 ± 3.22	16.66 ± 3.78
Jacobi et al. [36]	1999	22	Prospective	Pseudoexfoliation Glaucoma	24	71.3 ± 6.1	32.0 ± 7.7	18.5 ± 1.7	18.4 ± 1.7	18.0 ± 1.3
Merkur et al. [29]	2001	21	Retrospective	Pseudoexfoliation Glaucoma	18	81.57 ± 5.37	16.14 ± 2.50		11.63 ± 2.20	13.83 ± 2.32
Shingleton et al. [37]	2008	51	Retrospective	Pseudoexfoliation Glaucoma	60	78.2 ± 7.0	17.3 ± 5.2		15.8 ± 4.3	

**Table 2 jcm-13-00508-t002:** Author, year of publication, number of eyes, average difference with relative SD at 6, 12, and 24 months, study type and glaucoma subtype for the chosen studies as presented from the authors.

Paper	Year	N° of Eyes	Diff 6 Months	SD	Diff 12 Months	SD	Diff 24 Months	SD	Study Design	Glaucoma Subtype
Azuara-Blanco et al. [4]	2016	208	13.8	8.46	13.6	8.18	12.5	8.35	Prospective	Angle Closure Glaucoma
Dias-Santos et al. [5]	2014	15					5.4	7.81	Prospective	Angle Closure Glaucoma
El Sayed et al. [6]	2019	32	9.2	9.04	9	9.05			Prospective	Angle Closure Glaucoma
Hayashi et al. [7]	2001	74	6.1	4.32	6.1	4.75	6.9	4.23	Prospective	Angle Closure Glaucoma
Lai et al. [8]	2006	21					4.2	6.53	Prospective	Angle Closure Glaucoma
Lee et al. [9]	2009	48	4.2	2.77	4.3	2.65	4.5	2.98	Retrospective	Angle Closure Glaucoma
Moghimi et al. [10]	2015	46	8.4	6.64	8.3	6.64			Prospective	Angle Closure Glaucoma
Siak et al. [11]	2016	24	2.1	3.54	2	3.06			Prospective	Angle Closure Glaucoma
Tham et al. [12]	2008	35	1.6	3.67	2	3.73	1.8	3.85	Prospective	Angle Closure Glaucoma
Tham et al. [13]	2009	27	8.1	6.4	8.9	6.3	8.3	6.63	Prospective	Angle Closure Glaucoma
Tham et al. [14]	2013	26	8.2	5.25	9.6	5.72	8.4	6.55	Prospective	Angle Closure Glaucoma
Cetinkaya et al. [15]	2015	112	3.67	4.68	0.96	4.68	0.25	1.39	Prospective	Ocular Hypertension
Mansberg et al. [16]	2012	63	3.73	4.52	3.48	4.32	3.25	3.75	Retrospective	Ocular Hypertension
Poley et al. [17]	2008	81					5.4	3.42	Retrospective	Ocular Hypertension
Tanito et al. [18]	2001	36	1.7	2.46	1	1.92	0.3	1.6	Prospective	Ocular Hypertension
Anders et al. [19]	1997	42	3.64	4.47	3.71	4.55			Prospective	Open Angle Glaucoma
Arthur et al. [20]	2014	37	2.1	5.09	2.5	5.09	2.1	5.45	Retrospective	Open Angle Glaucoma
Craven et al. [21]	2012	123			0.9	3.86	0.1	3.99	Prospective	Open Angle Glaucoma
Damji et al. [22]	2006	29	1.54	5.49	1.67	4.55	1.54	4.92	Prospective	Open Angle Glaucoma
Fea et al. [23]	2015	24			1.1	2.98			Prospective	Open Angle Glaucoma
Francis et al. [24]	2014	80	0.2	4.12	0.6	4.2	0.8	3.9	Prospective	Open Angle Glaucoma
Gimbel et al. [25]	1995	53	3.7	3.37	4.2	3.37	3.8	3.17	Prospective	Open Angle Glaucoma
Hayashi et al. [7]	2001	68	4.6	6.19	4.4	6.14	5.5	5.95	Prospective	Open Angle Glaucoma
Iancu et al. [26]	2014	38	2.8	2.61	2.2	3.09			Prospective	Open Angle Glaucoma
Lee et al. [9]	2009	48	3	2.79	3.2	2.47	4	2.32	Retrospective	Open Angle Glaucoma
Liaska et al. [27]	2014	31					2.75	3.5	Prospective	Open Angle Glaucoma
Mathalone et al. [28]	2005	58			1.5	5.88	1.2	4.99	Retrospective	Open Angle Glaucoma
Merkur et al. [29]	2001	23	2.32	4.24	1.89	3.51			Retrospective	Open Angle Glaucoma
Pfeiffer et al. [30]	2015	50			9.2	5.01	7.4	5.64	Prospective	Open Angle Glaucoma
Samuelson et al. [31]	2011	117			1	3.95			Prospective	Open Angle Glaucoma
Samuelson et al. [32]	2018	187					5.3	4.47	Prospective	Open Angle Glaucoma
Shoji et al. [33]	2007	35	2	3.07	0.9	4.19	1.2	4.43	Retrospective	Open Angle Glaucoma
Siak et al. [11]	2016	30	1.6	3.56	2.2	4.06			Prospective	Open Angle Glaucoma
Siegel et al. [34]	2015	52	1.7	4.17	1.5	4.07	2.7	4.76	Retrospective	Open Angle Glaucoma
Vold et al. [35]	2016	131			6.2	4.34	5.4	4.42	Prospective	Open Angle Glaucoma
Damji et al. [22]	2006	29	4.08	3.71	3.23	3.88	3.17	4.28	Prospective	Pseudoexfoliation Glaucoma
Jacobi et al. [36]	1999	22	13.5	7.55	13.6	7.55	14	7.55	Prospective	Pseudoexfoliation Glaucoma
Merkur et al. [29]	2001	21			4.51	2.98	2.31	3.05	Retrospective	Pseudoexfoliation Glaucoma
Shingleton et al. [37]	2008	51			1.5	2.36			Retrospective	Pseudoexfoliation Glaucoma

## Data Availability

No new data were created or analyzed in this study. Data sharing is not applicable to this article.

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
