# Peer review of "Does Cataract Extraction Significantly Affect Intraocular Pressure of Glaucomatous/Hypertensive Eyes? Meta-Analysis of Literature"

_jcm, 2024, doi:10.3390/jcm13020508_

Round 1

Reviewer 1 Report

Comments and Suggestions for Authors

While the authors reviewed those studies involving phacoemulsification cataract surgery, can they comment on whether there would be any differences in IOP study results if manual small incision cataract surgery (MSICS) was undertaken instead?

Comments on the Quality of English Language

Mild grammatical edits recommended.

Author Response

  1. While the authors reviewed those studies involving phacoemulsification cataract surgery, can they comment on whether there would be any differences in IOP study results if manual small incision cataract surgery (MSICS) was undertaken instead?

Answer: You suggested commenting on whether there would be differences if manual small incision cataract surgery was performed instead of phacoemulsification. We found this topic quite interesting, and we implemented it in the text because phacoemulsification is not available in all parts of the globe. According to Sengupta et al., 2016, one of the more relevant studies on this matter, there are no statistical differences in IOP reduction between the two surgical techniques at 6 months after surgery, even though glaucomatous patients were excluded from the study; it will probably be matter of future debates the difference of the effect in these patients. [lines 280-284 tracked version]

Reviewer 2 Report

Comments and Suggestions for Authors

Dear author,

the manuscript becomes more valuable if consider the following points:

1- the title is inconsistent.

2- why did the author choose 6,12 and 24-month intervals?

3. are the significant decline in IOP after cataract in all groups

4. why choose both retro and prospective studies?

with regards,

Comments on the Quality of English Language

Dear Editor,

the overall is of average merit.

Author Response

  1. Dear author, the manuscript becomes more valuable if consider the following points:

1- the title is inconsistent.

2- why did the author choose 6,12 and 24-month intervals?

  1. are the significant decline in IOP after cataract in all groups
  2. why choose both retro and prospective studies?

Answer:  You expressed doubts about the consistency of the title with the article and we do agree that it can be considered too generic therefore we decided to replace it with: “Does cataract extraction significantly affect intraocular pressure of glaucomatous/hypertensive eyes? A metanalysis of literature”, to remain pertinent to the text and address the main topic of the analysis.

Regarding your second observation, these predetermined time spans are partially chosen in an arbitrary manner and partially are dictated by follow-up times from the considered studies.

About the third observation, as presented by the Forest-plots, the reduction is statistically significant at all the timepoints with a sufficient number of studies to perform the meta-analysis (6, 12, and 24 months from baseline for OAG and AGC, 12 months from baseline for PXG, and 24 months from baseline for OH. The main purpose of the analysis is evaluating the persistence of an effect often considered transient.

About the last issue, we decided to include both prospective and retrospective studies to aggregate as many high-quality studies as possible to obtain a more “real-world practice” adherent analysis, despite it increasing the heterogeneity. We accounted for that using a random-effects model.

Reviewer 3 Report

Comments and Suggestions for Authors

Cataract extraction surgery is one of the best in terms of cost/benefit analysis and still is the most performed surgical procedure every year in several countries. The objective of this meta-analysis was demonstrating the correlation between cataract surgery and the evolution of the most crucial and the only modifiable factor in glaucoma, Intraocular Pressure. I consider this study to have valuable data that would be of interest if published. As ophthalmic surgeon, I am glad that the results of this big meta-analysis confirmed my everyday observation of beneficial effect of cataract surgery on IOP reduction in glaucoma patients.

Author Response

  1. Cataract extraction surgery is one of the best in terms of cost/benefit analysis and still is the most performed surgical procedure every year in several countries. The objective of this meta-analysis was demonstrating the correlation between cataract surgery and the evolution of the most crucial and the only modifiable factor in glaucoma, Intraocular Pressure. I consider this study to have valuable data that would be of interest if published. As ophthalmic surgeon, I am glad that the results of this big meta-analysis confirmed my everyday observation of beneficial effect of cataract surgery on IOP reduction in glaucoma patients.

Answer:  We deeply thank you for your praises to our work.

Reviewer 4 Report

Comments and Suggestions for Authors

The article is well written, anyway to have an "easy reading article I suggest to condense the Literature research method from row 96 to 116 (could be all deleted or condensed). The flow diagram is very clear. Also analysis and data synthesis could be condensed maybe from line 131 to 154 could be deleted.

Comments on the Quality of English Language

English language should be reviewed by a native english teacher there are some "latin translated expressions"  

Author Response

  1. The article is well written, anyway to have an "easy reading article I suggest to condense the Literature research method from row 96 to 116 (could be all deleted or condensed). The flow diagram is very clear. Also analysis and data synthesis could be condensed maybe from line 131 to 154 could be deleted.

Answer:  We agree that the “analysis and data synthesis” section is not easy to read, so we decided to trim the redundant parts and the lines describing the “fixed-effects model” that was not used for the analysis due to high heterogeneity. We also find it relevant to explain the reasons why we excluded certain studies even if they initially seemed to respect the inclusion and exclusion criteria.